# Teachers' Autonomy-Supportive Behaviour and Learning Strategies Applied by Students: The Role of Students' Growth Mindset and Classroom Management in Low-SES-Context Schools

**Agne Brandisauskiene** [1], **Loreta Buksnyte-Marmiene** [2,*], **Ausra Daugirdiene** [1], **Jurate Cesnaviciene** [3], **Gabija Jarasiunaite-Fedosejeva** [2], **Egle Kemeryte-Ivanauskiene** [1] and **Rasa Nedzinskaite-Maciuniene** [1]

1   Educational Research Institute, Education Academy, Vytautas Magnus University, K. Donelaičio Str. 58, LT-44248 Kaunas, Lithuania; agne.brandisauskiene@vdu.lt (A.B.); ausra.daugirdiene@vdu.lt (A.D.); egle.kemeryte-ivanauskiene@vdu.lt (E.K.-I.); rasa.nedzinskaite-maciuniene@vdu.lt (R.N.-M.)
2   Department of Psychology, Faculty of Social Sciences, Vytautas Magnus University, Jonavos Str. 66, LT-44191 Kaunas, Lithuania; gabija.jarasiunaite-fedosejeva@vdu.lt
3   Teacher Training Institute, Education Academy, Vytautas Magnus University, K. Donelaičio Str. 58, LT-44248 Kaunas, Lithuania; jurate.cesnaviciene@vdu.lt
*   Correspondence: loreta.buksnyte-marmiene@vdu.lt

**Abstract:** This study explored a moderated mediation model, which is based on the assumption about the direct relationship between teachers' autonomy-supportive behaviours and students' learning strategies and via perceived classroom management as well as the effect of growth mindset for the relationships between (1) teachers' autonomy-supportive behaviours and learning strategies applied by students and (2) perceived classroom management and learning strategies applied by students. Data were collected from 23 secondary schools in municipalities in Lithuania with low SES (socioeconomic status) contexts (N = 581 students). The results showed that teachers' autonomy-supportive behaviours are directly and positively related to the increased use of learning strategies applied by the student. The perceived classroom management mediates the relationship between teachers' autonomy-supportive behaviours and learning strategies applied by the students. In this case, we have an inconsistent mediation, so the mediator acts as a suppressor (the indirect path through mediator is negative, while the direct is positive). The growth mindset does not moderate the relationship between perceived teachers' autonomy supportive behaviours and the students' use of learning strategies, but growth mindset moderates the relationship between perceived classroom management and learning strategies applied by the student. For students with a lower or moderate growth mindset, greater perceived classroom management is related to the lesser use of learning strategies. However, for those with a higher growth mindset, greater perceived classroom management is related to greater use of learning strategies. This study contributes to the understanding of the importance of teachers' behaviours for students' learning by focusing on classroom management and growth mindset.

**Keywords:** teachers' autonomy-supportive behaviours; learning strategies; growth mindset; classroom management; low SES context

## 1. Introduction

The new global education agenda widens the focus on skills development in schools and work to acquire knowledge, skills, values and attitudes that promote citizenship, resilience, empathy, tolerance, sustainability and peace [1]. Therefore, cross-cutting competencies such as problem solving, critical thinking, strategic competences, self-awareness and so on are necessary for all learners of all ages worldwide [2]. In other words, the goal of educating self-regulated learners to become full citizens of society is a challenge for all education systems.

While pursuing this goal, it is important to live up to the promise of the 2030 Agenda for Sustainable Development—no one should be left out. More specifically, the fourth Sustainable Development Goal (SDG4) seeks to guarantee access to inclusive, equitable quality education and lifelong learning opportunities for all, especially those in vulnerable situations or of other status [1]. International studies on students' achievement (e.g., OECD PISA) show that one of the most vulnerable groups is students from low socio-economic (SES) contexts. As in other countries (e.g., Israel, Luxembourg, Germany, Hungary, etc.), in Lithuania, students with high SES achieve better results than their peers with low SES [3,4]. It can be seen that the impact of personal background circumstances on student performance is partly mediated by other factors, e.g., students' access to educational resources, differences in opportunities to learn, grade repetition, etc. Jensen [5] claims that the major factor affecting the achievement of students living in unfavourable conditions is not their living environment, but rather the school and the teachers. Therefore, the behaviour of teachers during the teaching process is especially important, as they can strengthen or inhibit the active functioning of students, determining whether they can become self-regulated learners.

Thus, the analysis of the learning process and teacher–student interactions within it has shown that, in addition to educational factors (such as curriculum and pedagogy), psychological factors [6] can play a significant role in influencing student's behaviour and achievement, such as teacher motivational style and student's growth mindset.

Hence, this research seeks to determine whether students' perceived teacher autonomy-supportive behaviour encourages them to use learning strategies more actively. In order to achieve this goal, the constructs of students' growth mindset, classroom management and teacher motivational style (behaviour), as well as learning strategies and contextual factors, are explained and discussed in the theoretical background section. In the methodology section, specific information regarding the sample and its characteristics, measurement instruments, data collection and analysis procedure are explained. The results section is arranged according to the order of the research questions. In the final sections (discussion and conclusion), the results of our study are discussed in relation to previous studies, and final conclusions are presented.

## 2. Theoretical Background

The theoretical basis of this work is self-determination theory [7,8], which helps to understand the role of the teacher in the learning process of students. A teacher's motivational style, which can be either autonomy-supportive or a highly controlling style [9], may encourage or inhibit students' intrinsic motivational resources and engagement in learning. If a teacher adopts behaviours that support student autonomy, they respond to the three basic psychological needs of students (the needs for competence, relatedness and autonomy) and enhance students' development across an array of cognitive, personal and social indicators [7]. Such students will tend to set learning goals for themselves, engage more and more actively in deep and meaningful learning activities, rely more on their intrinsic motivational resources and strive for excellence [8,10,11]. Research shows that a self-sustaining environment creates a space that encourages greater student engagement in learning and increases intrinsic motivation, self-confidence and well-being [12–15].

According to researchers [14,16], all students, regardless of their gender, socioeconomic status or cultural background, have a tendency to a natural growth trend (intrinsic motivation, curiosity and autonomy), which forms the basis for self-regulated learning. However, Hornstra et al. [17] show that teachers find it harder to teach at-risk students in autonomy-supportive ways. In this case, teachers may be less likely to support students' needs for competence, relatedness and autonomy and not to recognize students' feelings, nor to provide them with appropriate help, information and choice, thus creating an environment that restricts students' natural psychological needs. Recognizing that teachers' behaviours are an important factor in student learning, we hypothesize that teachers'

autonomy-supportive behaviours enable students to make more active use of a variety of learning strategies.

The scientific literature identifies many factors associated with academic success, and one of them is learning strategies [18–21]. Learning strategies are the various actions that students take to improve their learning, such as actively repeating learning materials, asking complex questions, linking new information to what is already known, understanding information, drawing diagrams or explaining to other students. Learning strategies include a deep understanding of the learning material, linking it to existing prior knowledge and meaningful practice of the new learning material (knowledge, skills) over time [22]. The meaningful use of learning strategies helps students to better control their learning, and this increases their self-confidence and motivation to learn. According to Dembo and Seli [23], learning strategies are one of six components (along with motivation, time management, social environment, physical environment and control of performance) that students need to control if they want to learn successfully. The ability to use effective learning strategies is associated with a greater ability to acquire knowledge, understand it and, accordingly, be prepared to apply it in a variety of contexts.

Research shows that students from low social backgrounds do not make sufficient use of learning strategies [19]. According to researchers [18,20], encouraging these students to use a variety of learning strategies could improve their learning process, reduce the incidence of misbehaviour, and help them overcome various challenges at school. On the other hand, it turns out that another important factor influencing learning outcomes is the academic expectations placed on these students. It has been found that families [24] and teachers [25] tend to have lower academic expectations of children from a low SES environment than their peers from a higher SES environment. In this case, it is likely that these students have less confidence in the development of their abilities, put less effort in their learning and put less effort into adapting and using a variety of learning strategies. In other words, teachers may transmit their beliefs to students and may shape students' mindsets [26].

The implicit intelligence theory states that people can be categorized according to their implicit beliefs about ability [27,28]. Students with a growth mindset believe they can develop abilities through effort, learning and their own hard work. The growth mindset is the belief that intellectual ability can be developed. The fixed mindset is the opposite of the growth mindset. Students with a fixed mindset believe that abilities are innate and cannot be developed. Growth and fixed mindsets affect students' behaviours and how they deal with challenges. Those with a fixed mindset tend to avoid challenges, are cautious about taking risks outside their comfort zone, may lack initiative, give up easily when things become difficult, fear making mistakes and become defensive when failure occurs. They see failure as a result of their lack of ability. However, students with a growth mindset boldly accept challenges, see efforts as a way to master something, are more creative and understand that mistakes help them learn and grow [29]. Thus, the growth mindset is associated with higher resilience and academic achievement among students facing difficulty [30]. Students with a growth mindset are motivated to learn and make an effort by practicing, in seeking alternate strategies and in seeking the help of others needed for their progress [31]. Therefore, we hypothesize that students with a more pronounced growth mindset will be more active in applying learning strategies than those with a fixed mindset.

Analysing low SES students, it was observed that they are more characterized by fixed mindset than growth mindset [32]. It could be assumed that a fixed mindset is a risk factor in the learning process of low SES students, preventing them from revealing and exploiting their full potential, and a growth mindset is a protective factor that reduces the negative impact of low SES on the student learning process [33]. However, research on this issue is mixed: some studies confirm a link between growth mindset and low SES student achievement [33], while others suggest a nonsignificant relationship between growth mindset and learning among lower SES students [34]. Moreover, although research

on the relationship between student growth mindset and their achievement is available, there is a lack of research that analyses the role of student growth mindset in their learning-related behaviours, such as the use of learning strategies.

Finally, teachers' conceptions of student learning are also related to classroom management, and this phenomenon (classroom management) is related to students' autonomy support [35]. According to researchers [36], classroom management is one of the basic conditions for student learning and a central aspect of effective teaching. However, research on this issue is relatively small [35,37], and the studies presented often refer to only a few general teaching principles [38]. Classroom management involves all the actions teachers take to maintain a smooth and productive classroom environment so that students can learn effectively. The teacher can apply both preventive and reactive strategies to create the right learning environment. Research suggests that proper classroom management is associated with greater student focus, a favourable learning environment and student engagement in the learning process [35,39].

It should be noted that classroom rules and teacher interventions to achieve proper order in the classroom are the visible side of classroom management. However, researchers [40,41] point out that factors that help manage the classroom include student–teacher relationships, care for students or mindset. Thus, it becomes clear that class management has at least two dimensions. The first can be described as the demarcation of certain teacher behaviours and the maintenance of discipline (often, this can be linked to control), and the second can be described as the development of favourable relationships with students and a favourable mental set as a preventive strategy to prevent student misconduct, which is associated with the expression of care. Both of these components are important because control is necessary for managing student misbehaviour, and care itself may not be enough to manage misbehaviour well [37]. Additionally, research results [37] suggest that classrooms with more students from low SES show more manifestations of misbehaviour, and a certain teacher's behaviour control is necessary in order to create an orderly and learning-favourable environment. However, in classrooms where students are from high SES and there are few manifestations of misbehaviour, students may perceive teacher control as unnecessary and overly strict, which can lead to negative emotional reactions in students. Thus, different strategies can be used to manage a classroom, and they can affect students differently, so it is important to explain how classroom management relates to a teacher's autonomy-supportive behaviour, students' growth mindset and the learning strategies applied by students.

In summary, it can be seen that student's abilities to actively apply various learning strategies in the learning process is an important condition for successful learning. It is therefore important to understand what factors encourage the use of active learning strategies. We believe that teachers' autonomy-supportive behaviours, as such, increase students' engagement in learning activities, promote the expression of their independence and competencies and significantly increase the use of active learning strategies by students. However, teachers' autonomy-supportive behaviours alone are not enough for the learning process to be successful. Another important factor in this process is class management, which is defined as teachers' ability to manage student behaviour and disruptions and to set classroom rules and norms that make classroom processes efficient [42]. Both teachers' autonomy-supportive behaviour and class management are seen as important for students' growth mindset development and growth mindset culture in the classroom [26,29], i.e., they promote students' growth mindset expression. We believe that students' growth mindset, which encourages students' personal effort and belief of the possibility to develop abilities, moderates the interaction between teacher behaviour and the learning strategies used by the student.

We hypothesized that perceived teachers' autonomy-supportive behaviours are directly related to students' use of learning strategies and through a mediator—perceived classroom management. Additionally, growth mindset moderates the relationship between perceived teachers' autonomy-supportive behaviour and students' use of learning strate-

gies as well as perceived classroom management and the use of learning strategies, so that the relationship is stronger for students with higher rather than lower growth mindsets. A diagram of the conceptual model is presented in Figure 1.

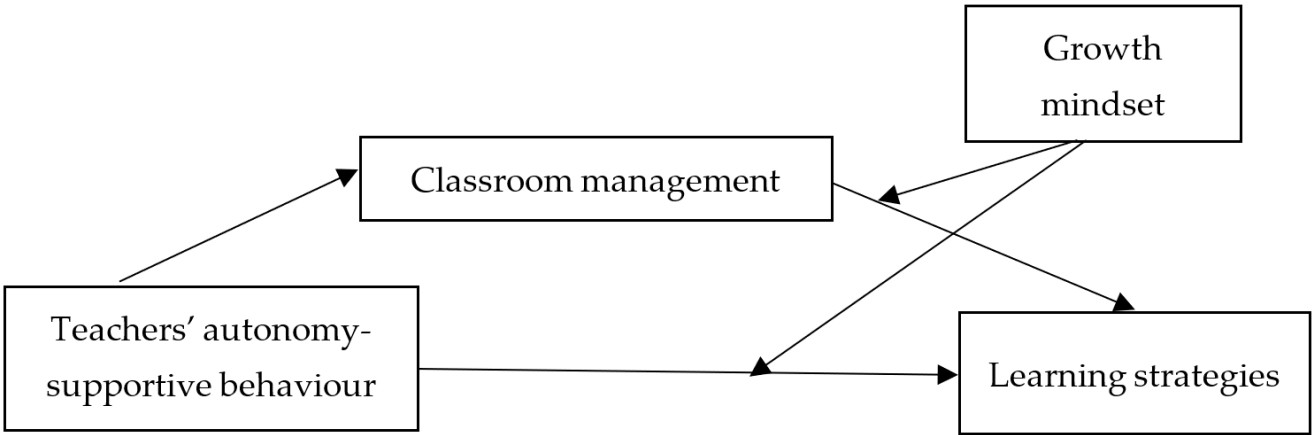

**Figure 1.** A diagram of the conceptual model.

## 3. Materials and Methods

### 3.1. Sample and Data Collection Procedure

The study was conducted using a quantitative research design in May 2021. It was approved by the Research Ethics Committee of the Education Academy of Vytautas Magnus University (protocol number SA-EK-21-03). Additionally, the research procedures were carried out according to the Declaration of Helsinki.

In total, 23 general education schools from municipalities of Lithuania with low SES contexts were purposefully selected for the study. In terms of number of students, these schools were in small towns or villages. A significant number of students (from 24.4% to 34.3%) studied were from low-income families. The principals of the selected schools were first contacted, and permission was received to conduct the study. Further, both students and their parents/caregivers received a consent form before participation so that they were able to select whether they consented to respond to the survey. Only students whose parents gave written consent voluntarily participated in the study. The anonymity and confidentiality of study data were guaranteed to study participants. Students completed a self-report questionnaire on the online platform at their convenience at home at https://apklausa.lt/ (accessed on 31 May 2021).

The sample in the present study included 581 students from 7th to 10th grade (Table 1).

**Table 1.** Sample characteristics.

|  | **Boys** | | **Girls** | | **Total** | |
|---|---|---|---|---|---|---|
|  | **n** | **%** | **n** | **%** | **n** | **%** |
| 7th grade | 55 | 20.0 | 81 | 26.5 | 136 | 23.4 |
| 8th grade | 59 | 21.5 | 66 | 21.5 | 125 | 21.5 |
| 9th grade | 74 | 26.9 | 73 | 23.9 | 147 | 25.3 |
| 10th grade | 87 | 31.6 | 86 | 28.1 | 173 | 29.8 |
| Total | 275 | 100 | 306 | 100 | 581 | 100 |

### 3.2. Measurement

Students' perceived teachers' autonomy-supportive behaviours were measured by the short form of the *Learning Climate Questionnaire* (LCQ) [43]. The questionnaire, consisting of six items, was assessed through a 7-point Likert-type scale ranging from 1 (strongly disagree) to 7 (strongly agree). The Learning Climate Questionnaire was downloaded

from the website https://selfdeterminationtheory.org (accessed on 8 February 2021). Written permission to use this measure was obtained by the first author of this article. One sample item was "My teacher provides me with choices and options". The scores on the 6-item were calculated by averaging the individual item scores. Higher scores indicated stronger perceived autonomy support. The LCQ in this study showed evidence of reliability (Cronbach's $\alpha$ = 0.926).

Learning strategies applied by students were assessed using *Learning strategies' scale* (statements for the scale were formulated by the first and third authors of the article). After considering several key factors—the age of our research subjects, their SES environment and the multifaceted nature of our study (many variables)—the decision was made to create our own learning strategies' questionnaire that would be concise and understandable for our research subjects. This scale consisted of five items answered on a 5-point Likert-type scale from 1 (almost never) to 5 (almost always). One example of an item is: "I use a variety of techniques (such as repeating orally, doing diagrams) to memorize the information I need." Scores were calculated by averaging the five items, where higher scores reflect a more active application of learning strategies. Scores were calculated by averaging the five items, where higher scores reflect a more active application of learning strategies. The reliability of the learning strategies scale, as assessed by Cronbach's alpha, was 0.847.

Classroom management was rated on a scale of 5 statements drawn up by the first author of this article. The sample items were "Students don't listen to what the teacher is saying" and "There's noise during the lesson." The items were scored on a 5-point Likert-type scale from 1 (almost never) to 5 (almost always). Total scores were created by reverse-scoring all the items and averaging ratings from each item, with higher scores representing better class management. Cronbach's alpha shows that reliability of the scale is sufficient (0.852).

The *Growth mindset questionnaire* contained eight statements drawn up by the first author of the article. We had to create this in accordance with research ethics because we did not receive any response from other researchers as to whether we could use a corresponding growth mindset scale. Four statements intended to establish a growth mindset (e.g., No matter what my capabilities are, I can always change them a little); the other four were for a fixed mindset (e.g., I have certain capabilities and I cannot change them). Answers to each statement were presented on a 4-point Likert scale from 1 (strongly disagree) to 4 (strongly agree). Prior to data analysis, the responses to the fixed-minded subscale statements were transcoded in reverse. Total scores were calculated by averaging students' responses to the four growth and fixed items, respectively. Lower scores indicated a fixed mindset (interpreting test results described as lower growth mindset) and higher scores a growth mindset (interpreting test results described as higher growth mindset). The Cronbach's $\alpha$ reliability coefficients for *Growth mindset questionnaire* were 0.580.

As covariates, gender (1 = boys, 2 = girls) and grade (1 = 7th and 8th grade, 2 = 9th and 10th grade), physical activity and sleep sufficiency were included in the study model.

The *Physical Activity Questionnaire for Adolescents* (PAQ-A) [44] was used to measure the general levels of physical activity. Questions encompassed different activities leading to harder breathing or an increase in heart rate, during a wide range of periods (namely physical education class, school breaks, lunch times, after school, in the evenings, on weekends and in general during last week). For example: "In the last 7 days, during your Physical Education classes, how often were you very active (playing hard, running, jumping throwing)?" Students respond on a 5-point Likert-type scale (the lowest activity response being 1, and the highest activity response being 5). The total score of the PAQ-A is generated from the mean of all questions, with higher scores indicating more frequent participation in physical activity. Cronbach's alpha test showed the PAQ-A reached acceptable reliability, $\alpha$ = 0.898.

Sleep sufficiency was assessed by three questions: *What time do you usually go to bed on school days? What time do you usually go to bed on weekends? What time do you usually wake up on school days?* Bedtimes ranged in intervals from "No later than 9 p.m." to "2 a.m. or later".

Wake times ranged in half-hour intervals from "No later than 5 a.m." to "8 a.m. or later". Having evaluated self-reported sleep duration and bedtime schedules, sleep duration was calculated and dichotomized as ≥8 h of sleep (i.e., sufficient sleep) and <8 h of sleep (i.e., insufficient sleep). Based on this, students were divided into two groups: those who do not sleep enough (1) and those who sleep enough (2).

### 3.3. Common Method Bias

The data collection instrument for this study was a self-report questionnaire. The dependent and independent variables were measured within one survey, using the same response method (ordinal scales). Therefore, we checked and controlled for common method bias. Following the suggestion by Podsakoff, MacKenzie, Lee, and Podsakoff [45], we ran the Harman's single factor test with an unrotated factor solution in an exploratory factor analysis. The results showed that the first factor could only account for 25.13% of the variance, which was less than the critical value of 40%, indicating that the common method bias was not significant.

### 3.4. Statistical Analysis

The data analysis was performed using IBM Statistical Package for the Social Sciences (IBM SPSS) version 23.0 and Hayes' PROCESS macro (version 3.5) for testing moderated mediation [46]. Normality tests showed that data related to the main study variables (perceived teachers' autonomy-supportive behaviours, learning strategies applied by the student and perceived classroom management and growth mindset) were close to a normal distribution, based on the assessment of skewness and kurtosis for large samples [47,48]. Thus, Pearson correlation analysis was used to test the relationships between all main study variables before analysing moderated mediation.

When testing the proposed model, all independent variables were mean-centred to alleviate multicollinearity and improve the interpretation of regression coefficients [49]. Gender, grade, physical activity and sleep sufficiency were included in the model as covariates because analysis showed that they were related to the main study variables. To ensure that the results were not affected by multivariate outliers, Mahalanobis distance estimates were calculated for the predictors of the moderated mediation model and excluded from the analysis.

Bootstrap samples of 10,000 and a confidence interval of 95% were selected for the analysis. The chosen statistical significance level (*p*-value) was 0.05. $R^2$ change (extra variance explained) was calculated to obtain the moderation effect and the effect size of a model. To explore the effects of moderator in moderated mediation, simple slope tests were calculated; significant interactions were also plotted. Indices of the indirect effect of moderated mediation were considered statistically significant if the 95% CI, estimated using the bootstrap method, did not include zero.

## 4. Results

### 4.1. Descriptive Statistics and Primary Analysis

Means, standard deviations and Pearson's correlations of the dependent, independent variables, mediator and moderator were calculated (see Table 2).

As expected, the use of learning strategies applied by students positively correlated with perceived teachers' autonomy-supportive behaviours, while perceived classroom management positively correlated with perceived teachers' autonomy-supportive behaviours. Additionally, growth mindset was positively associated with perceived teachers' autonomy-supportive behaviours, students' use of learning strategies and perceived classroom management. All bivariate correlations were statistically significant (*p* < 0.001) but weak or moderate (0.1 < r < 0.5) [50].

**Table 2.** Descriptive statistics and correlations among main study variables.

| | M | SD | 1 | 2 | 3 | 4 |
|---|---|---|---|---|---|---|
| 1. Teachers' autonomy-supportive behaviours | 4.45 | 1.437 | 1 | | | |
| 2. Learning strategies | 3.12 | 0.791 | 0.496 *** | 1 | | |
| 3. Classroom management | 3.61 | 0.833 | 0.154 *** | 0.042 | 1 | |
| 4. Growth mindset | 2.85 | 0.407 | 0.313 *** | 0.339 *** | 0.234 *** | 1 |

Note: *** $p < 0.001$.

### 4.2. Moderated Mediation Effects

The results of the moderated mediation model are presented in Figure 2. The results showed the significant direct positive effect of perceived teachers' autonomy-supportive behaviours to the students' use of learning strategies ($\beta = 0.217$, S.E. = 0.020, CI = [0.178; 0.257], $p < 0.001$). As perceived teachers' autonomy-supportive behaviours increase, the use of learning strategies applied by students also increases. Additionally, perceived classroom management mediates the relationship between teachers' autonomy-supportive behaviours and the learning strategies applied by the students. As perceived teachers' autonomy-supportive behaviours increase, so does perceived classroom management ($\beta = 0.080$, S.E. = 0.025, CI = [0.032; 0.128], $p < 0.05$). However, as perceived classroom management increases, students' use of learning strategies decreases ($\beta = -0.075$, S.E. = 0.034, CI = [$-0.141$; $-0.010$], $p < 0.05$). In this case, we have an inconsistent mediation, so the mediator acts as a suppressor (the indirect path through mediator is negative, while the direct is positive) (see Figure 2 and Table 3).

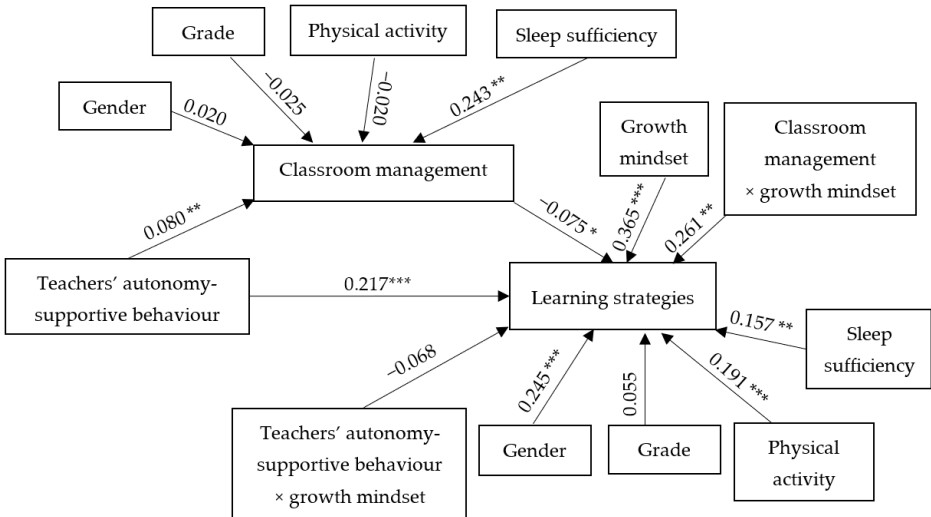

**Figure 2.** The importance of perceived classroom management and growth mindset for the relationship between perceived teachers' autonomy-supportive behaviours and learning strategies applied by the students. Notes: *** $p < 0.001$; ** $p < 0.01$; * $p < 0.05$.

With reference to the results, growth mindset has a positive direct relationship with the use of learning strategies applied by students. The higher growth mindset, the greater the use of learning strategies applied by the students ($\beta = 0.365$, S.E. = 0.070, CI = [0.227; 0.503], $p < 0.001$). Moreover, growth mindset moderates the relationship between perceived classroom management and learning strategies applied by the students ($\beta = 0.261$, S.E. = 0.082, CI = [0.099; 0.422], $p < 0.001$). However, growth mindset does not moderate the relationship between perceived teachers' autonomy-supportive behaviours and the students' use of learning strategies ($\beta = -0.068$, S.E. = 0.043, CI = [$-0.152$; 0.016], $p > 0.05$).

**Table 3.** The importance of perceived classroom management and growth mindset for the relationship between perceived teachers' autonomy-supportive behaviours and learning strategies applied by students. Statistically significant differences are written in bold format.

| Predictors | β | Est./S.E. | t | 95% CI | p Value |
|---|---|---|---|---|---|
| | | | **Classroom Management** | | |
| Teachers' autonomy-supportive behaviours | 0.080 | 0.025 | 3.282 | [0.032; 0.128] | **0.001** |
| Gender | 0.020 | 0.068 | 0.297 | [−0.114; 0.155] | 0.766 |
| Grade | −0.025 | 0.069 | −0.368 | [−0.161; 0.110] | 0.713 |
| Physical activity | −0.020 | 0.053 | −0.376 | [−0.124; 0.084] | 0.707 |
| Sleep sufficiency | 0.243 | 0.071 | 3.443 | [0.104; 0.382] | **0.001** |
| F | | | 5.889 | | |
| R$^2$ | | | 0.049 | | |
| | | | **Learning Strategies Applied by the Student** | | |
| Teachers' autonomy-supportive behaviours | 0.217 | 0.020 | 10.858 | [0.178; 0.257] | **<0.001** |
| Classroom management | −0.075 | 0.034 | −2.250 | [−0.141; −0.010] | **0.025** |
| Growth mindset | 0.365 | 0.070 | 5.193 | [0.227; 0.503] | **0.001** |
| Teachers' autonomy-supportive behaviours × growth mindset | −0.068 | 0.043 | −1.588 | [−0.152; 0.016] | 0.113 |
| Classroom management × growth mindset | 0.261 | 0.082 | 3.172 | [0.099; 0.422] | **<0.001** |
| Gender | 0.245 | 0.054 | 4.562 | [0.140; 0.351] | **<0.001** |
| Grade | 0.055 | 0.054 | 1.013 | [−0.051; 0.161] | 0.312 |
| Physical activity | 0.191 | 0.041 | 4.607 | [0.109; 0.272] | **<0.001** |
| Sleep | 0.157 | 0.056 | 2.815 | [0.047; 0.266] | **0.005** |
| F | | | 36.242 | | |
| R$^2$ | | | 0.364 | | |

Notes: CI—Confidence Interval; 95% CI also presented for unstandardised coefficients.

Simple slope tests indicated that the relationship between perceived classroom management and the students' use of learning strategies changes direction depending on whether a student has a higher or lower growth mindset. For students with a lower or moderate growth mindset, greater perceived classroom management is related to the lesser use of learning strategies. However, for those with a higher growth mindset, greater perceived classroom management is related to the greater use of learning strategies (see Figure 3). The effect size of interaction is small ($\Delta$R$^2$ = 0.0112) though significant ($p$ = 0.002).

Conditional direct and indirect effects of perceived teachers' autonomy-supportive behaviours on learning strategies applied by students for different moderator values are presented in Table 4. As we can see from the table, the direct effects of perceived teachers' autonomy-supportive behaviours on learning strategies applied by the students are stronger than the indirect effects. Additionally, conditional direct effects describing the relationship between perceived teachers' autonomy-supportive behaviours and the students' use of learning strategies are positive, while conditional indirect effects describing the relationship through a mediator perceived classroom management are negative (except for describing the relationship for students with a lower growth mindset).

Even though a growth mindset does not moderate the relationship between perceived teachers' autonomy-supportive behaviours and the use of learning strategies applied by students (the moderator effect is insignificant), we can see that the described direct relationship is stronger for the students with lower and moderate growth mindsets than for those with a higher growth mindset. Considering conditional indirect effects, for students with a lower growth mindset, the indirect relationship is stronger than for those with moderate or higher growth mindset. It is also worth mentioning that the indirect growth

mindset effect for the group of students with a higher growth mindset is insignificant when considering the confidence intervals. The index of moderated mediation is 0.021.

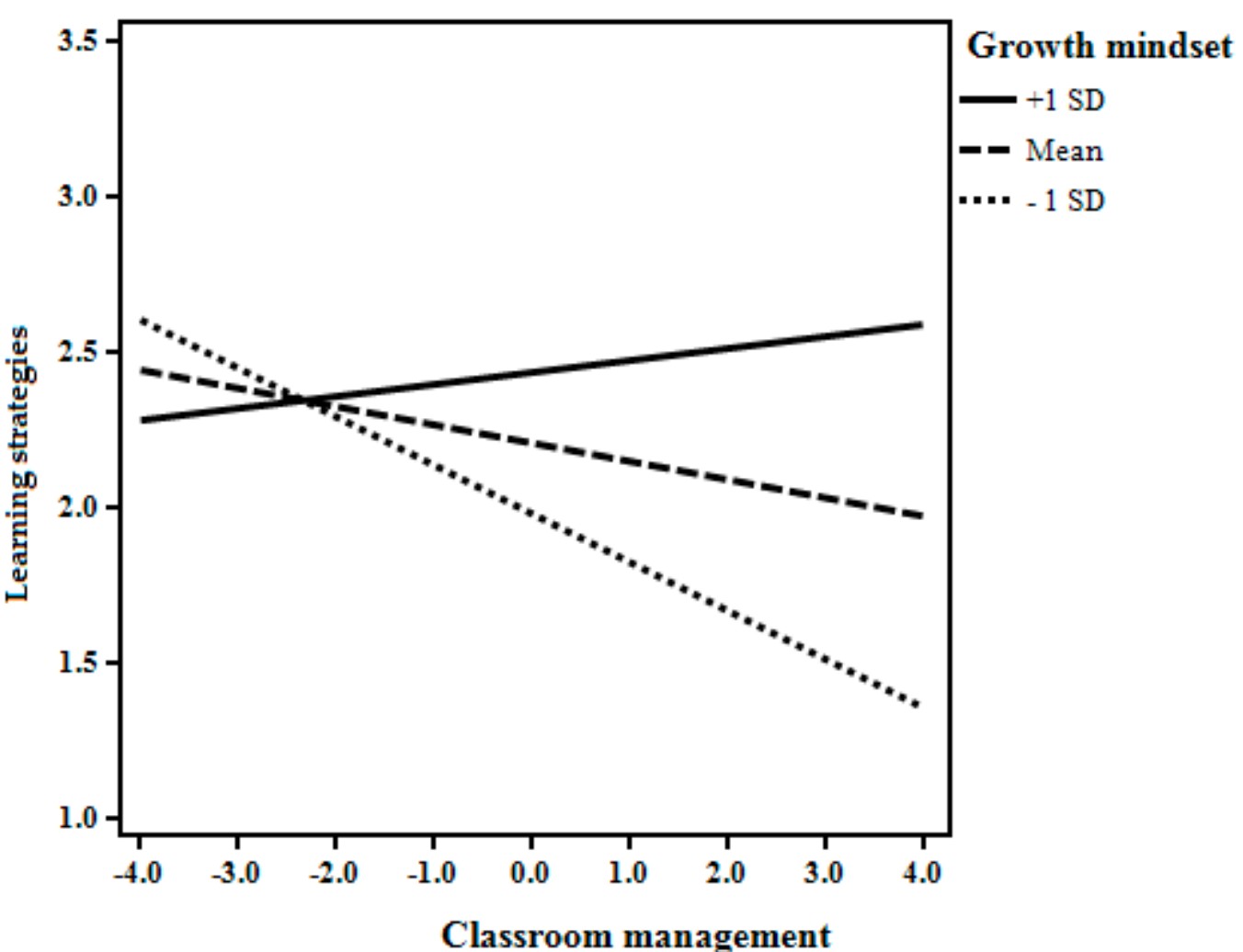

**Figure 3.** The effect of growth mindset for the relationship between classroom management and learning strategies.

**Table 4.** The conditional direct and indirect effects of teachers' autonomy-supportive behaviours on learning strategies applied by the student for different moderator values.

| | Effect | SE | CI |
|---|---|---|---|
| | **Direct Effects** | | |
| Higher growth mindset | 0.190 | 0.028 | [0.136; 0.244] |
| Moderate growth mindset | 0.217 | 0.020 | [0.178; 0.257] |
| Lower growth mindset | 0.245 | 0.025 | [0.195; 0.294] |
| | **Effect** | **BootSE** | **BootCI** |
| | **Indirect Effects** | | |
| Higher growth mindset | 0.002 | 0.004 | [−0.005; 0.011] |
| Moderate growth mindset | −0.006 | 0.003 | [−0.013; −0.001] |
| Lower growth mindset | −0.015 | 0.006 | [−0.028; −0.004] |
| | *Index Of Moderated Mediation* | | |
| | 0.021 | | |

Note: CI—Confidence Interval.

## 5. Discussion

As already mentioned in the theoretical part of the article, this study relied on self-determination theory [7,8], which reveals that when a teacher promotes students' autonomy, they become more involved in the learning process [12–15]. The study confirms this statement and our hypothesis, that perceived teachers' autonomy-supportive behaviours are positively related to the application of learning strategies; that is, as perceived teachers' autonomy-supportive behaviours increase, students are more active in applying learning strategies. The results show a significant direct positive effect of perceived teachers' autonomy-supportive behaviours on the learning strategies applied by students. Empirical findings from other researchers [51,52] have also confirmed that students become more self-regulated learners when their learning climate is autonomy-supportive. Interestingly, however, we can observe from the model of conditional processes that this positive relationship is stronger for students with lower or moderate growth mindsets than for those with higher ones. The results obtained are thought-provoking and raise the question of how to interpret such results. We believe that these results should not imply that teachers' autonomy-supportive behaviours are worthless for students with high growth mindsets (research shows that teachers' autonomy-supportive behaviours encourage all students to use learning strategies more actively, which is beneficial), but for those students with lower growth mindsets, such teacher behaviour is particularly important because it is likely to encourage the student to interpret such teacher behaviour as belief in their abilities. Children growing up in an unfavourable environment are dominated by feelings of helplessness, and they tend to attribute learning failures to external factors [29]. Teachers' autonomy-supportive behaviours encourage students to take personal responsibility, strengthen the sense of control over the situation and ability to change the results and stimulate students' self-confidence, which is contrary to their sense of helplessness. Similar observations are described by other researchers, arguing that such teaching allows students to experience greater self-confidence and competence need satisfaction that energizes their challenge-seeking, classroom engagement, skill development, performance and the use of deep and sophisticated learning strategies [53]. Thus, teachers' autonomy-supportive behaviours are congruent with growth mindset intervention, which suggests that students' success is due to their personal effort, self-control and hard work and thus leads to more positive outcomes [29]; in our study, this relates to a more active use of learning strategies.

The direct relationship between teachers' autonomy-supportive behaviours and learning strategies applied by students was found to be much stronger than indirect (through a mediator). By introducing a classroom management variable as a mediator, it shows the value of this variable for the relationship being analysed. There is a significant indirect relationship between these constructs (behaviours that promote teacher autonomy and the application of student learning strategies) through classroom management. The more students that perceive teachers' behaviours as supporting autonomy, the more likely they are to feel that the teacher manages the classroom. Reeve and Cheon [54] state that teachers' autonomy-supportive behaviours are associated with a better classroom climate, so students may feel that the teacher is in better control of the classroom. However, greater classroom management is associated with the less active use of learning strategies. Assessing the whole indirect relationship, the results of the study show that classroom management inhibits (weakens) the relationship between maintaining perceived teachers' autonomy and applying learning strategies. This contradicts the scholars' assertion that autonomy support together with teachers' guidance and structure make the learning process the most effective [26,55]. This established (admittedly rather unexpected) result of the study raises several assumptions. First, the results of our study suggest that the phenomenon of classroom management is multifaceted, and that the different strategies used by a teacher may affect students differently [37], because, as researchers state [35], classroom management is a continuum from control to autonomy support. Second, we would like to point out that our study of classroom management was assessed by students and analysed only one component of classroom management—students' perceived environment in terms

of discipline when a student perceives the conditions for learning in the classroom as appropriate or inappropriate. Clearly, the results obtained do not allow us to examine what behavioural strategies the teacher applied to maintain discipline in the classroom (this has not been studied). Strategies that inhibit student activity may have been used, which are sometimes misunderstood as class management methods. We recognize this as a limitation of this study and would like to note that in further research, it is very important to delve into teacher behaviour strategies, as research [e.g., 56] shows that teachers with effective classroom management skills are aware of the high needs of students and have specific techniques to meet these needs. Marzano [56] suggests specific strategies for those students who fear failure in the learning process, as well as different techniques for perfectionist students, and so on. Finally, we would like to draw attention once again to the context of our study. There are quite a few students from low SES (up to 34%) among the students surveyed, but in accordance with research ethics, we did not identify them. Studies by other researchers suggest that these students may be characterized by a fixed (lower) growth mindset [57] and may have more behavioural problems in the classroom [37], so we hypothesize that a research result (indirect relationship, that classroom management inhibits (weakens) the link between perceived teachers' autonomy support and applying learning strategies) may be related to the specificity of this study sample.

The results of our study show that the relationship between perceived classroom management and the learning strategies applied by students changes direction depending on whether a student has a higher or a lower growth mindset. For students with a lower or moderate growth mindset, greater perceived classroom management is related to the less active use of learning strategies. However, for those with a higher growth mindset, greater perceived classroom management is related to the more active use of learning strategies. However, the negative indirect relationship through class management (although it is weaker than the direct one) is also stronger among those with a lower growth mindset than among those with an average or higher one. Thus, it becomes clear that in the model we have developed, classroom management is a phenomenon that does not provide unambiguous results and has a different impact on students with higher or lower growth mindsets. Further research should continue to deepen our knowledge of preventive and reactive classroom management strategies [37] and constructs [40] that can create a favourable learning environment for students from low SES.

We must also recognize that a student's growth mindset is a very relevant factor in the learning process [26,29], especially in terms of self-regulation in the process of learning. Research shows that students with a growth mindset are more likely to use self-regulation strategies [58] and are more likely to invest more effort and change strategies when faced with challenges [59] than those who believe that intelligence is fixed. On the one hand, from our point of view, the result of the study is very significant—that with a more pronounced growth mindset, students are more active in using learning strategies. Consequently, a student's belief in the development of their abilities, which is inherent in a higher growth mindset, encourages the student to put in more personal effort (more active use of learning strategies) to assimilate the material needed to learn. The obtained correlations probably reflect the characteristic of students with a high growth mindset—not to give up in the face of difficulties and to look for a solution to the situation. The obtained research results could motivate teachers to apply interventions aimed at students' growth mindset development, to implement growth mindset culture in the classroom as encouraging more active involvement of students in the learning process [42]. Teachers' autonomy-supportive practices may create an important context for students' growth mindset development [26]. On the other hand, a growth mindset does not moderate the relationship between a teacher's perceived autonomy-supportive behaviour and the application of learning strategies. However, the relationship between autonomy-supportive behaviour and the application of learning strategies is positive, as the perceived teachers' autonomy-supportive behaviours increase, so does the application of learning strategies. From the conditional process model, we can observe that this positive relationship is

stronger among students with lower or moderate growth mindsets than those with higher ones, meaning that behaviours that support student autonomy are important for students with lower growth mindsets. These findings from our study bring us back to the assumption made by other researchers [26,60] that students' mindset may be affected by contextual variables.

Finally, the results of the study reveal the importance of individual characteristics of the student, such as age, gender, sleep and physical activity, as contextual variables that affect the learning process of students. The results of the study show that girls tend to be more active in applying learning strategies than boys. This confirms data from other researchers that girls are more involved in the learning process than boys [61,62]. Our study also suggests that more physically active students are more active in using learning strategies. This finding is supported by other studies where researchers show that more physically active students are more engaged in their classroom lessons and are positively associated with cognitive (processing speed, working memory, planning) engagement [63], and also that physical activity-induced positive emotions lead to broadened thoughts and behaviours and facilitate more adaptive responses (such as problem solving and seeking assistance). Physical activity also increases adolescents' responsibility, self-regulated learning and prosocial behaviour (such us taking turns, asking for permission to speak, congratulating others, following rules and handling or dealing with conflicts) [64]. The data from the study suggest that classroom management and learning strategies are also related to adolescent sleep duration. Various studies confirm the unequivocal significance of sleep during the period of adolescence. Students' adequate sleep is associated with a learning-friendly, disciplined classroom environment, while lack of sleep can affect students' cognitive processes [65], especially memory, attention and emotion processing [66], and promote adolescent opposition behaviour, hyperactivity, risk-taking behaviour [67] and delaying and skipping lessons [68,69].

In summary, the results of the study suggest that in schools with a high number of students from low SES backgrounds, the teacher's autonomy-supportive behaviour is significant for the active use of learning strategies by students. The presented model is statistically significant and explains 41.3 percent variances in the use of learning strategies. Students with a more pronounced growth mindset tend to be more active in using learning strategies in an appropriate, disciplined classroom environment. We would therefore like to emphasize the practical value of this study: teachers must engage in autonomy-supportive behaviour when working with low SES students. It is noteworthy that, according to researchers [54,70], this style of motivation is plastic and can be learned by teachers. Additionally, our study reaffirms the importance of the growth mindset. Teachers should strive to help students believe that skills are being developed through effort. We believe that the teacher can create the conditions for effective learning, that is, the use of learning strategies, by applying autonomy-supportive behaviour and strengthening students' perseverance while performing learning tasks. However, we recognize that classroom management and growth mindsets are multifaceted phenomena whose interrelationships and interactions need to be further explored. We used our own Learning Strategies Questionnaire in this study, but it would be appropriate to continue the research using other questionnaires designed to investigate this object. This would provide an opportunity to learn more about the analysed educational phenomena and perhaps gain new insights. Such research would provide better answers to questions about how to create a learning-friendly environment for all students, regardless of their gender, age or SES. It would also serve the fourth Sustainable Development Goal, looking for the appropriate and effective educational practices, taking into account different groups of students and recognizing an inclusive approach, because students are very different.

## 6. Conclusions

This study provides insight into perceived teachers' autonomy-supportive behaviours as directly related to the learning strategies applied by students and through a mediator—

perceived classroom management. In this case, we have inconsistent mediation, so the mediator acts as a suppressor (the indirect path through mediator is negative, while the direct is positive). The growth mindset does not moderate the relationship between perceived teachers' autonomy supportive behaviours and the learning strategies applied by students, but the growth mindset moderates the relationship between perceived classroom management and learning strategies applied by the student. For students with a lower or moderate growth mindset, greater perceived classroom management is related to the lesser use of learning strategies. However, for those with a higher growth mindset, greater perceived classroom management is related to the greater use of learning strategies. The results obtained contribute to the understanding of the importance of teachers' autonomy-supportive behaviours for students' learning strategies and calls for further research on classroom management and growth mindset, which are heterogeneous phenomena.

**Author Contributions:** Conceptualization, A.B., J.C., A.D., E.K.-I., L.B.-M. and R.N.-M.; writing—A.B., L.B.-M. and R.N.-M.; data collecting J.C. and E.K.-I.; data analysis, interpretation of the results J.C., A.D. and G.J.-F.; review and editing, A.B., A.D., E.K.-I, G.J.-F. and L.B.-M. All authors have read and agreed to the published version of the manuscript.

**Funding:** This research was funded by the Research Council of Lithuania (LMTLT) project "Creating a Supportive Learning Environment: in Search of Factors enabling the School Community" (Agreement No. S-DNR-20-1), co-financed by the European Union under the measure "Implementation of Analysis and Diagnostics of Short-term (Necessary) Research (in Health, Social and Other Fields) related to COVID-19".

**Institutional Review Board Statement:** The study was conducted according to the guidelines of the Declaration of Helsinki and approved by the Ethics Committee of Education Academy Vytautas Magnus University, Lithuania (Protocol number: SA-EK-21-03).

**Informed Consent Statement:** Informed consent was obtained from all subjects involved in the study.

**Data Availability Statement:** The datasets generated and analysed during the current study are not publicly available due to privacy and ethical concerns but are available from the corresponding author on reasonable request.

**Conflicts of Interest:** The authors declare no conflict of interest.

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
