# Peer review of "Teachers’ Autonomy-Supportive Behaviour and Learning Strategies Applied by Students: The Role of Students’ Growth Mindset and Classroom Management in Low-SES-Context Schools"

_sustainability, doi:10.3390/su14137664_

Round 1
Reviewer 1 Report
The article seems an interesting one as it investigates mediators and moderators of the relationships between important educational variables. Some comments to improve follow.
Usually, we do not build questionnaires if we have already such questionnaiers. The authors should clarify why they wrote the learning strategies questionnaire and not used an existing one.
The preivious comment applies to the Growth mindset questionnaire.
The authors should not only comment on the directin of the correlation in table 2, but also on the strength of those correlations, which are in part very low.
Figure 2 is not clear. The authors should clarify it. For examples why some variables appear in it more than once?
The conclusions summerize the results. The authors need to elaborate in detail how we can utilize the results. In addtion, they should mention the limitations of the research.
Author Response
We would like to thank a lot for the time that the Editor and the reviewers have spent on reading our manuscript and provided meaningful suggestions to improve it further. Your comments allowed us to take a new look at our research and deepened our experience in writing manuscript. We appreciate your efforts to make our manuscript better. Thank you very much. The changes which we made in the manuscript regarding reviewers’ comments are highlighted in the paper by using blue colored text.
Below are our responses to the reviewer’ comments:

Reviewer 2 Report
The quality of work is excellent. It has a correct theoretical argument. The methodology used is rigorous and uses the appropriate procedures and instruments.
The results are correctly presented and analyzed rigorously and thoroughly.
The discussion and the conclusions are very revealing and this research is very interesting for the scientific community.
However, the Discussion section can be expanded with comparison of previous studies on this topic.
Author Response
We would like to thank a lot for the time that the Editor and the reviewers have spent on reading our manuscript and provided meaningful suggestions to improve it further. Your comments allowed us to take a new look at our research and deepened our experience in writing manuscript. We appreciate your efforts to make our manuscript better. Thank you very much. The changes which we made in the manuscript regarding reviewers’ comments are highlighted in the paper by using blue colored text.
Below are our responses to the reviewer’ comments:
|
Reviewer’s 2 comment |
Authors’ answer |
|
The discussion and the conclusions are very revealing and this research is very interesting for the scientific community. However, the Discussion section can be expanded with comparison of previous studies on this topic. |
We are thankful for Your comment. We supplemented the Discussion section regarding Your comment (see manuscript, lines 441-443; 458-461; 473-475;484-485; 521-524; 544-546). |
Reviewer 3 Report
It seemed to me a manuscript with a topic of high interest, well written and well founded.
As the only change I would suggest the authors modify Figure 1 to remove the crossed arrows.
Author Response
We would like to thank a lot for the time that the Editor and the reviewers have spent on reading our manuscript and provided meaningful suggestions to improve it further. Your comments allowed us to take a new look at our research and deepened our experience in writing manuscript. We appreciate your efforts to make our manuscript better. Thank you very much. The changes which we made in the manuscript regarding reviewers’ comments are highlighted in the paper by using blue colored text.
Below are our responses to the reviewer’ comments:
|
Reviewer’s 3 comment |
Authors’ answer |
|
As the only change I would suggest the authors modify Figure 1 to remove the crossed arrows.
|
We appreciate for Your comment. We want to clarify that we cannot remove intersecting lines because it is a theoretical model of research. Moderation was tested in two places/positions, in direct connection as well as in mediation.
|
Round 2
Reviewer 1 Report
You need to clarify the questionnaire issue in order for the reader to understand why you had to build a new questionnaire. You need to refer to this in the methodology section and in the recommendations section as well.
Author Response
We would like to thank a lot for the time that the Editor and the reviewers have spent on reading our manuscript and provided meaningful suggestions to improve it further.
We supplemented the text in the sections Measurements and Discussion regarding Your comment (see manuscript, lines 253-257; 272-274; 590-593).
The changes which we made in the manuscript regarding Reviewer's comments are marked up by using yellow colored text and the “Track Changes” function.
Round 3
Reviewer 1 Report
Thanks for the amendments.